# Fact-driven Logical Reasoning

## Abstract

Logical reasoning deeply relies on accurate, clearly presented clue forms which are usually modeled as entity-like knowledge in existing studies. However, in real hierarchical reasoning motivated machine reading comprehension (MRC), such one-side modeling are insufficient for those indispensable local complete facts or events when only "global" knowledge is really paid attention to. Thus, in view of language being a complete knowledge/clue carrier, we propose a general formalism to support representing logic units by extracting backbone constituents of the sentence such as the subject-verb-object formed "facts", covering both global and local knowledge pieces that are necessary as the basis for logical reasoning. Beyond building the ad-hoc graphs, we propose a more general and convenient fact-driven approach to construct a supergraph on top of our newly defined fact units, and enhance the supergraph with further explicit guidance of local question and option interactions. Experiments on two challenging logical reasoning MRC benchmarks show that our proposed model, FOCAL REASONER, outperforms the baseline models dramatically.

## 1 Introduction

Machine reading comprehension (MRC) requires machine to answer question according to given passage [1, 2, 2, 3, 4]. Logical reasoning [5] from MRC accounts for human intuition about entailment of sentences and reflects the semantic relations between sentential constituents [6]. Recently, there is a surging trend of research into logical reasoning ability, among which ReClor [7] and LogiQA [5] are two representative datasets introduced to promote the development of logical reasoning, where logical reasoning questions are selected from standardized exams such as GMAT[1], requiring models to read and comprehend the complicated logical relationships. Similar to the standard question-answering (QA)-based MRC tasks in form, our concerned logical reasoning QA tasks contain three elements: passage, question and the candidate options as examples shown in Figure 1.

MRC models usually exploit a pre-trained language model (PrLM) as a key encoder for effective contextualized representation. Meanwhile, the major challenge of logical reasoning is to uncover logical structures, and reasoning with the candidate options and questions to predict the correct answer. However, it is difficult for PrLMs to capture the logical structure inherent in the texts since logical supervision is rarely available during pre-training. Existing logical reasoning has shown serious dependence on knowledge-like clues. This is due to the lengthy, noisy text in human language which is though a natural carrier of knowledge but does not provide a clean, exact knowledge form. Thus, an increasing interest is using graph networks to model the entity-aware relationships in the passages [8, 9, 10, 11]. However, all these methods may insufficiently capture indispensable logical units from two perspectives. First, they mostly focus on entity-aware commonsense knowledge, but pay little attention to those non-entity, non-commonsense clues [12]. Second, when existing models

---

[1]`https://en.wikipedia.org/wiki/Graduate_Management_Admission_Test`

Submitted to 35th Conference on Neural Information Processing Systems (NeurIPS 2021). Do not distribute.

| Question | Passage | Answer |
|---|---|---|
| Example 1

From this we know | Xiao Wang is taller than Xiao Li,
Xiao Zhao is taller than Xiao Qian,
Xiao Li is shorter than Xiao Sun, and
Xiao Sun is shorter than Xiao Qian. | ✓ A. Xiao Li is shorter than Xiao Zhao.
B. Xiao Wang is taller than Xiao Zhao.
C. Xiao Sun is shorter than Xiao Wang.
D. Xiao Sun is taller than Xiao Zhao. |
| Example 2

Which one of the follow-
ing statements, most seriously
weakens the argument? | .... A large enough comet colliding
with Earth could have caused a cloud
of dust that enshrouded the planet
and cooled the climate long enough
to result in the dinosaurs' demise. | A. Many other animal species from same era did not
become extinct at the same time the dinosaurs did.
B. It cannot be determined from dinosaur skeletons whether
the animals died from the effects of a dust cloud.
C. The consequences for vegetation and animals of a comet
colliding with Earth are not fully understood.
✓ D. Various species of animals from the same era and similar
to them in habitat and physiology did not become extinct. |

Figure 1: Two examples from LogiQA and ReClor respectively are illustrated. There are arguments and relations between arguments. Both are emphasized by different colors: arguments, relations. Key words in questions are highlighted in Purple. Key options are highlighted in gray.

extract predicate logic inside language into knowledge, they only exploit quite limited predicates like *hasA* and *isA* but ignore a broad range of predicates in real language. From either of the perspectives, the existing methods actually only concern about those "global" knowledge that keeps valid across the entire data, without sufficient "local" perception of complete facts or events in the given specific part of MRC task. We argue such insufficient modeling on logic units roots from the ignorance of language itself being the complete knowledge/clue carrier. Thus, we propose extracting a kind of broad *facts* according to backbone constituents of a sentence to effectively cover such indispensable logic reasoning basis, filling the gap of local, non-commonsense, non-entity, or even non-knowledge clues in existing methods as shown in Figure 2. For example, these units may reflect the facts of *who did what to whom*, or *who is what* in Figure 3. Such groups can be defined as "fact unit" following [13] in Definition 1. The fact units are further organized into a supergraph following Definition 2.

**Definition 1** *(Fact Unit) Given an triplet $T = \{E_1, P, E_2\}$, where $E_1$ and $E_2$ are arguments (including entity and non-entity), $P$ is the predicate between them, a fact unit $F$ is the set of all entities in $T$ and their corresponding relations.*

**Definition 2** *(Supergraph) A supergraph is a structure made of fact units (regarded as subgraphs) as the vertices, and the relations between fact units as undirected edges.*

As shown in Figure 2, we regard the defined *fact* as the results of syntactic processing, rather than those from semantic role labeling (SRL) as in previous study, thus the proposed *fact* also extends the processing means in existing work. Correspondingly, in this work, we propose a fact-driven logical reasoning model, called FOCAL REASONER, which builds supergraphs on top of fact units as the basis for logical reasoning, to capture both global connections between facts and the local concepts or actions inside the fact. In addition, we strengthen our model by the question-option-aware interaction. Specifically, we explicitly reformulate questions with negation expressions to compensate for the

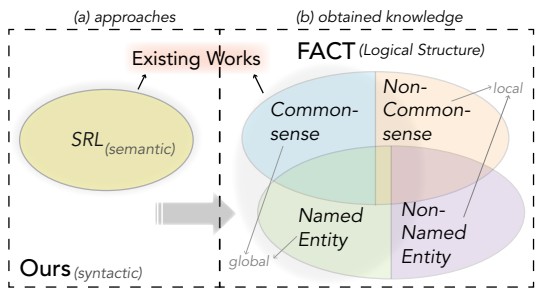

Figure 2: Our "fact" V.S. existing approaches.

insensitiveness of PrLMs, all of which are interacted in our supergraph. Such resulted FOCAL REASONER is evaluated on two challenging logical reasoning benchmarks including ReClor, LogiQA, and one dialogue reasoning dataset Mutual for generalizability, achieving new state-of-the-art results.

## 2 Related Work

**Machine Reading Comprehension**  Recent years have witnessed massive researches on Machine Reading Comprehension, which has become one of the most important areas of NLP [14, 15, 16,

17, 18, 19, 20, 21, 22]. Despite the success of MRC models on various datasets such as CNN/Daily Mail [1], SQuAD [2], RACE [3] and so on, researchers began to rethink to what extent does the problem been solved. Nowadays, there are massive researches into the reasoning ability of machines. According to [23, 24, 25], reasoning abilities can be broadly categorized into (1) commonsense reasoning [26, 27, 28, 29]; (2) numerical reasoning [30]; (3) multi-hop reasoning [31] and (4) logical reasoning [5, 7], among which logical reasoning is essential in human intelligence but has merely been delved into. Natural Language Inference (NLI) [32, 33, 34] is a task closely related to logical reasoning. However, it has two obvious drawbacks in measuring logical reasoning abilities. One is that it only has three logical types which are *entailment, contradiction* and *neutral*. The other is its limitation on sentence-level reasoning. Hence, it is important to research more comprehensive and deeper logical reasoning abilities.

**Logical Reasoning in MRC**   There are two main kinds of features in language data that would be the necessary basis for logical reasoning: 1) *knowledge*: global facts that keep consistency regardless of the context, such as commonsense, mostly derived from named entities; 2) *non-knowledge*: local facts or events that may be sensitive to the context, mostly derived from detailed language. Existing works have made progress in improving logical reasoning ability [8, 9, 10, 11, 12, 38]. However, these approaches are barely satisfactory as they mostly focus on the global facts such as typical entity or sentence-level relations, which are obviously not sufficient.   In this work, we strengthen the basis for logical reasoning by unifying both types of the features as "facts". Different from previous studies that focus on the knowledge components, we propose a fact-driven logical reasoning framework that builds supergraphs on top of fact units to capture both global connections between entity-aware facts and the local concepts or events inside the fact.

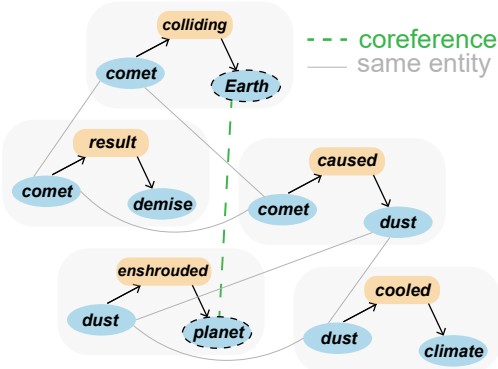

Figure 3: An example of constructed supergraph. In contrast, the dotted vertices and edges are focused in most existing studies [35, 36, 37].

## 3   Approaches

In this section, we will describe our method in detail. The overall architecture of the model is shown in Figure 4 . We first construct a supergraph from the raw text based on the fact units extracted. Then we conduct reasoning over the supergraph with question-option guided approaches to learn and update the features, which are further incorporated in answer prediction.

### 3.1   Supergraph Construction

Figure 5 illustrates our method for constructing a supergraph from raw text inputs. The first step is to obtain triplets that constitute a fact unit. To keep the framework generic, we use a fairly simple fact unit extractor based on the syntactic relations. Given a context consisting multiple sentences, we first conduct dependency parsing of each sentence. After that, we extract the subject, the predicate, and the object tokens to get the `"Argument-Predicate-Argument"` triplets corresponding to each sentence in the context.

With the obtained triplets, the fact units are organized in the form of Levi graph [39], which turns arguments and predicates all into nodes. An original fact unit is in the form of $F = (V, E, R)$, where $V$ is the set of the arguments, $E$ is the set of edges connected between arguments, and $R$ is the relations of each edge which are predicates here. The corresponding Levi graph is denoted as $F_l = (V_L, E_L, R_L)$ where $V_L = V \cup R$, which makes the originally directly connected arguments be intermediately connected via relations. As for $R_L$, previous works such as [40, 41] designed three types of edges $R_L = \{default, reverse, self\}$ to enhance information flow. Here in our settings,

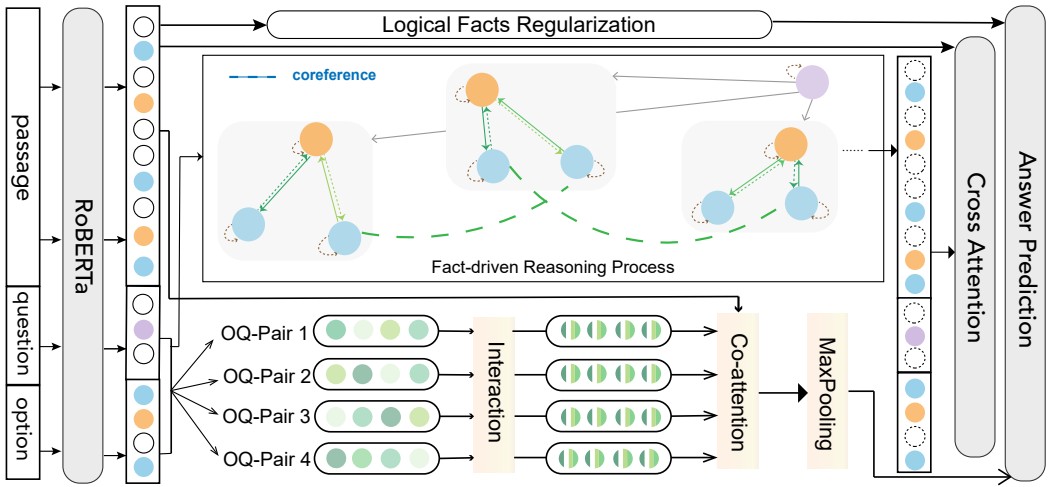

Figure 4: The framework or our model. For supergraph reasoning, in each iteration, each node selectively receives the message from the neighboring nodes to update its representation. The dashed circle means zero vector.

we extend it into five types: *default-in, default-out, reverse-in, reverse-out, self*, corresponding to the directions of edges towards the predicates.

We construct the supergraph by making connections between fact units $F_l$. In particular, we take three strategies according to question-option, identical concept and co-reference information. (1) For question-option pair, We initialize a global node $V_g$ with its representation and connect it to all the fact unit nodes. The edge type are set as *global*. The global node ensures that all fact units are connected so that information can be exchanged during graph encoding. (2) There can be identical mentions in different sentences, resulting in repeated nodes in fact units. We connect nodes corresponding to the same non-pronoun arguments by edges with edge type *same*. (3) We conduct co-reference resolution on context using an off-to-shelf model[2] in order to identify arguments in fact units that refer to the same one. We add edges with type *coref* between them. The final supergraph is denoted as $S = (F_l \cup V_g, E)$ where $E$ is the set of edges added with the previous three strategies.

## 3.2 Encoder

### 3.2.1 Context Encoder

Our context encoder $F_C(.)$ is initialized with a pre-trained language model, i.e., RoBERTa-large [42]. Question, context and option are concatenated and then fed into the encoder. If the question is detected to contain negative meanings, we add a special token <pos> before the question, else we add <neg>. In a whole, we get the hidden representation as following:

$$\{h_{c,0}, ..., h_{c,l_c+1}, h_{q,1}, ..., h_{o,1}, ..., h_{o,l_o+1}\} = F_C(\{x_{c,0}, ..., x_{c,l_c+1}, x_{q,0}, ..., x_{o,1}, ..., x_{o,l_o+1}\}), \tag{1}$$

where $x_{c,0} =$ , $x_{c,l_c+1} = x_{o,l_o+1} =$ , $x_{q,0} =$ <pos>/<neg> and $h_i \in \mathbb{R}^d$, $d$ is the hidden size.

### 3.2.2 Supegraph Encoder

**Graph Initialization**    $F_C(.)$ encodes each token in nodes $V_L$, and then the averaged hidden state is used as the initial representation of the original word of each node, because PrLMs like RoBERTa take subwords as input while our triplets extraction performs in word-level. For the global QA-context node, we averaged the embeddings of tokens in question and option for initialization. We also use a one-hot embedding layer to encode the relations between two nodes.

---

[2]https://github.com/huggingface/neuralcoref.

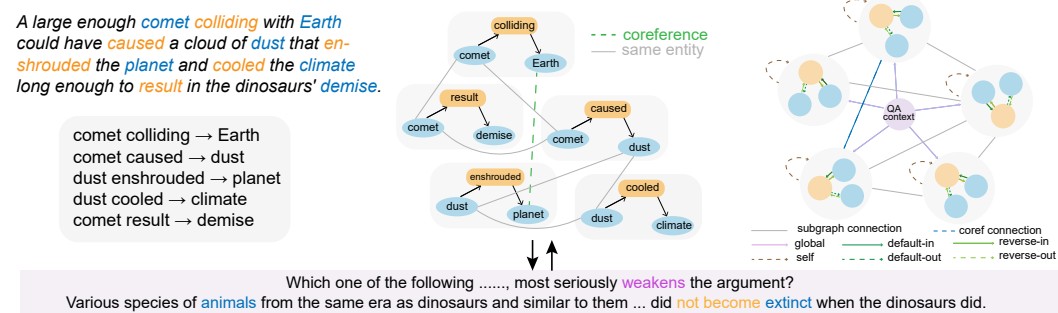

Figure 5: The process of constructing the fact chain and its corresponding Levi graph form of an example in Figure 1. Entities and relations are illustrated in its corresponding color.

**Graph Attention Network** Based on the relational graph convolutional network [43] and given the initial representation $h_i^0$ for every node $v_i$, the feed-forward or the message-passing process with information control can be written as:

$$h_i^{(l+1)} = \text{ReLU}(\sum_{r \in R_L} \sum_{v_j \in \mathcal{N}_r(v_i)} g_q^{(l)} \frac{1}{c_{i,r}} w_r^{(l)} h_j^{(l)}), \tag{2}$$

where $\mathcal{N}_r(v_i)$ denotes the neighbors of node $v_i$ under relation $r$ and $c_{i,r}$ is the number of those nodes. $w_r^{(l)}$ is the learnable parameters of layer $l$. $g_q^{(l)}$ is a gated value between 0 and 1.

Through the graph encoder $F_G(.)$, we then obtain the hidden representations of nodes in fact units as:

$$\{h_0^F, ...h_m^F\} = F_G(\{v_{L,0}, ...v_{L,m}\}, E_L). \tag{3}$$

These features are further concatenated to get the final node representation of the supergraph:

$$\{h_0^S, ...h_m^S\} = F_G(\{h_0^F, ...h_m^F\}, E_C). \tag{4}$$

For node features on the supergraph, it is fused via the attention and gating mechanisms with the original representations of the context encoder. Specifically, denoting the original whole sequence representation after context encoder as $H^C$, we apply attention mechanism to append the supergraph representation to the original one:

$$\tilde{H} = \text{Attn}(H^c, K_f, V_f), \tag{5}$$

where $\{K_f, V_f\}$ are packed from the learned representations of the supergraph. We compute $\lambda \in [0, 1]$ to weigh the expected importance of supergraph representation of each source word:

$$\lambda_1 = \sigma(W_\lambda \tilde{H} + U_\lambda H^C), \tag{6}$$

where $W_\lambda$ and $U_\lambda$ are learnable parameters. $H^C$ and $\tilde{H}$ are then fused for an effective representation:

$$H = H^C + \lambda \tilde{H} \in \mathbb{R}^{4 \times d}. \tag{7}$$

### 3.2.3 Question-Option-aware Interaction

Options have their inherent logical relations, which can be leveraged to aid answer prediction. Inspired by [44], we use an attention-based mechanism to gather option correlation information.

Specifically for an option $O_i$, the information it get by interaction with option $O_j$ is calculated as:

$$O_i^{(j)} = [O_i^q - O_i^q \text{Attn}(O_i^q, O_j^q; v); O_i^q \circ O_i^q \text{Attn}(O_i^q, O_j^q; v)], \tag{8}$$

where $O_i^q$ is the representation of the concatenation for the $i$-th option and question after the context encoder. Then the option-wise information are gathered to fuse the option correlation information:

$$\hat{O}_i = \tanh(W_c[O_i^q; \{O_i^{(j)}\}_{i \neq j}] + b_c), \tag{9}$$

where $\mathbf{W}_c \in \mathbb{R}^{d \times 7d}$ and $b_c \in \mathbb{R}^d$. Finally, a gating mechanism is used to fuse the option features:

$$O^q_{i,:k} = g_{i,:k} \circ O^q_{i,:k} + (1 - g_{i,:k}) \circ \hat{O}_{i,:k}, \tag{10}$$

where the $g_{i,:k} = \sigma(W_g[O_{i,:k}; O^{\hat{q}}_{i,:k}; \tilde{Q}] + b_g) \in \mathbb{R}^d$ is the $i$-th column of gate $g$.

## 3.3 Hierarchical Decoder

To better incorporate the information obtained above, apart from getting the original pooled context-attended representation $h^C \in \mathbb{R}^{4 \times d}$, we combine the attended vectors $O^f$ and $H$ from the previous encoder through a fusing layer.

$$\begin{aligned}
E_1 &= \mathrm{ReLU}(\mathrm{FC}([h^C, H, h^C - H, h^C \circ H])), \\
E_2 &= \mathrm{ReLU}(\mathrm{FC}([h^C, H, h^C - O^f, h^C \circ O^f])), \\
P &= \sigma(\mathrm{FC}([E_1, E_2])), \\
C &= P \circ H + (1 - P) \circ O^f \in \mathbb{R}^{4 \times d}.
\end{aligned} \tag{11}$$

Then another linear layer is applied for final prediction as $z = W_z C + b_z \in \mathbb{R}^4$. We seek to minimize the cross entropy loss over the correct decision $l$ by

$$\mathcal{L}_{ans} = -\log \mathrm{softmax}(z)_l. \tag{12}$$

**Logical Fact Regularization**  Inspired by [45], the embedding of the tail argument should be close to the embedding of the head argument plus a relation-related vector in the hidden representation space. Without loss of generality, we assume that in our settings, the summation of the subject vector and the relation vector should be close to the object vector as much as possible, i.e.,

$$v_{subject} + v_{relation} \rightarrow v_{object}. \tag{13}$$

In order to make the logical facts more of factual correctness, we introduce a regularization for the extracted logical facts based on the hidden states of the sequence $h_i$ where $i = 1, \dots, L$ and $L$ is the total length of the sequence. The regularization is defined as:

$$L_{lfr} = \sum_{k=1}^{m} (1 - \cos(h_{sub_k} + h_{rel_k}, h_{obj_k})), \tag{14}$$

where $m$ is the total number of logical fact triplets extracted from the context as well as the option and $k$ indicates the $k$-th fact triplet.

**Training Objective.**  During training, the overall loss for answer prediction is:

$$\mathcal{L} = \alpha \mathcal{L}_{ans} + \beta \mathcal{L}_{lfr}, \tag{15}$$

where $\alpha$ and $\beta$ are two parameters. In our implementation, we set $\alpha = 1.0$ and $\beta = 0.5$.

# 4 Experiments

## 4.1 Datasets

We conducted the experiments on three datasets. Two for specialized logical reasoning ability testing: ReClor [7] and LogiQA [5] and one for logical reasoning in dialogues: MuTual [46]. For more details, one can refer to Appendix A.

## 4.2 Implementation Details

We fine-tune RoBERTa as the backbone PrLM for FOCAL REASONER. The overall model is end-to-end trained and updated by Adam [47] optimizer with an overall learning rate 8e-6 for ReClor and LogiQA, and 4e-6 for MuTual. The weight decay is $0.01$. We set the warm-up proportion during training to $0.1$. Graph encoders are implemented using DGL, an open-source lib of python. The layer number of the graph encoder is 2 for ReClor and 3 for LogiQA. The maximum sequence length is 256 for LogiQA and MuTual, and 384 for ReClor. The model is trained for 10 epochs with a total batch size 16 and an overall dropout rate $0.1$ on 4 NVIDIA Tesla V100 GPUs, which takes around 2 hours for ReClor and 4 hours for LogiQA[3].

---

[3]Our code has been submitted along with this submission, which will be open after the blind review period.

| Model | ReClor | | | | LogiQA | |
|---|---|---|---|---|---|---|
| | Dev | Test | Test-E | Test-H | Dev | Test |
| Human [7] | - | 63.00 | 57.10 | 67.20 | - | 86.00 |
| BERT-Large [7] | 53.80 | 49.80 | 72.00 | 32.30 | 34.10 | 31.03 |
| XLNet-Large [7] | 62.00 | 56.00 | 75.70 | 40.50 | - | - |
| RoBERTa-Large [7] | 62.60 | 55.60 | 75.50 | 40.00 | 35.02 | 35.33 |
| DAGN [10] | 65.20 | 58.20 | 76.14 | 44.11 | 35.48 | 38.71 |
| DAGN (Aug) [10] | 65.80 | 58.30 | 75.91 | 44.46 | 36.87 | 39.32 |
| FOCAL REASONER | **66.80** | **58.90** | **77.05** | **44.64** | **41.01** | **40.25** |

Table 1: Experimental results of our model compared with baseline models on ReClor and LogiQA dataset. Test-E and Test-H denote Test-Easy and Test-Hard respectively. We performed Pitman's permutation test [48] and found that our model significantly outperformed the baseline (p<0.05).

| Model | MuTual | | | | | | $\text{MuTual}^{plus}$ | | | | | |
|---|---|---|---|---|---|---|---|---|---|---|---|---|
| | Dev Set | | | Test Set | | | Dev Set | | | Test Set | | |
| | $R_4@1$ | $R_4@2$ | MRR | $R_4@1$ | $R_4@2$ | MRR | $R_4@1$ | $R_4@2$ | MRR | $R_4@1$ | $R_4@2$ | MRR |
| RoBERTa$_{base}$ [46] | 69.5 | 87.8 | 82.4 | 71.3 | 89.2 | 83.6 | 62.2 | 85.3 | 78.2 | 62.6 | **86.6** | 78.7 |
| -MC [46] | 69.3 | 88.7 | 82.5 | 68.6 | 88.7 | 82.2 | 62.1 | 83.0 | 77.8 | 64.3 | 84.5 | 79.2 |
| FOCAL REASONER | **73.4** | **90.3** | **84.9** | **72.7** | **91.0** | **84.6** | **63.7** | **86.1** | **79.1** | **65.5** | 84.3 | **79.7** |

Table 2: Experimental results of our model compared with baseline PrLM on MuTual dataset.

## 4.3 Results

Tables 1 and 2 show the results on ReClor, LogiQA, and MuTual, respectively. All the best results are shown in bold. Based on our implemented baseline models (basically consistent with public results), we observe dramatic improvements on both of the logical reasoning benchmarks, e.g., on ReClor test set, FOCAL REASONER achieves $+4.2\%$ on dev set and $+3.3.\%$ on the test set. FOCAL REASONER also outperforms the prior best system DAGN[4], reaching $77.05\%$ on the EASY subset, and $44.64\%$ on the HARD subset. The performance suggests that FOCAL REASONER makes better use of logical structure inherent in the given context to perform reasoning than existing methods. On the dialogue reasoning dataset MuTual, our model achieves quite a jump compared with the RoBERTa-base LM[5]. This verifies our model's generalizability on other downstream reasoning task settings.

In addition, Table 5 lists the accuracy of our model on the dev set of ReClor of different question types. Results show that our model can perform well on most of the question types, especially "Strengthen" and "Weaken". This means that our model can well interpret the question type from the question statement and make the correct choice corresponding to the question.

## 5 Analysis

### 5.1 Ablation Study

To dive into the effectiveness of different components in FOCAL REASONER, we conduct an ablation study which takes RoBERTa as the backbone on the ReClor dev set. Table 3 summarizes the results.

**Supergraph reasoning:** The first key component is the supergraph reasoning. We ablate the global atom and erase all the edges connected with it. The results suggest that the global atom indeed betters message propagation, leveraging performance from $64.6\%$ to $66.8\%$. We also find that replacing the initial QA pair representation of the global atom with only question representation hurts the performance. In addition, without the logical fact regularization, the performance drops from $66.8\%$

---

[4]For a fair comparison, we only compare to public literatures with the same PrLM RoBERTa-large. The test results are from the official leaderboard https://eval.ai/web/challenges/challenge-page/503/leaderboard/1347.

[5]Since there are no official results on RoBERTa-large LM, we use RoBERTa-base LM instead for consistency.

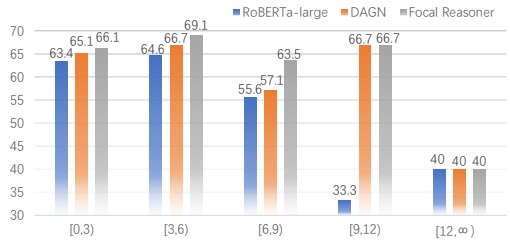 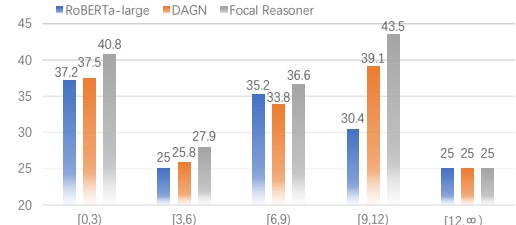

Figure 6: Accuracy of models on number of fact units on dev set of ReClor (left) and LogiQA (right).

to $64.2\%$, indicating its usefulness. For edge analysis, when (1) all edges are regarded as a single type rather than the original designed 8 types in total and (2) co-reference edges are removed, the accuracy drops to $63.7\%$ and $64.8\%$, respectively. It is proved that in our supergraph, edges link the fact units in reasonable manners, which properly uncovers the logical structures.

**Fact Units Variants** Apart from our syntactically constructed fact units, there are another two ways in different granularities for construction. We replace the fact units with named entities which are used in previous works like [49]. The statistics of fact units and named entities of ReClor and LogiQA are stated in Table 4, from which we can infer that there are indeed more fact units than named entities. Thus using fact units can better incorporate the logical information within the context. When replacing all the fact units with named entities, we can see from Table 3 that it significantly decreases the performance. We also explore the performance using semantic role labeling the similar way as in [50]. We can see that SRL, leveraging a much more complex information as well as computation complexity, fails to achieve a performance as good as our original fact unit.

**Interactions:** We further experimented with the query-option-interactions setting to see how it affects the performance. The results suggest that the features learned from the interaction process enhance the model. Considering that the logical relations between different options are a strong indicator of the right answer, this means that the model learns from a comparative reasoning strategy.

| Model | Accuracy |
|---|---|
| FOCAL REASONER | $66.8_{\pm 0.13}$ |
| **Supergraph Reasoning** | |
| - global node | $64.6_{\pm 0.32}$ |
| - co-reference | $64.8_{\pm 0.24}$ |
| - logical fact regularization | $64.2_{\pm 0.12}$ |
| - QA context node $\rightarrow$ Q node | $66.4_{\pm 0.16}$ |
| - question reformulation | $65.2_{\pm 0.16}$ |
| - edge type | $63.7_{\pm 0.19}$ |
| **Fact Unit Variants** | |
| - named entity | $62.8_{\pm 0.26}$ |
| - SRL | $62.2_{\pm 0.32}$ |
| **Interactions** | |
| - interactions | $65.5_{\pm 0.52}$ |

Table 3: Ablation results on the dev set of ReClor.

## 5.2 Effects of Fact Units Numbers

To inspect the effects of the number of fact units, we split the original dev set of ReClor and LogiQA into 5 subsets. The statistics of the fact unit distribution on the datasets are shown in Table 6. Numbers of fact units for most contexts in ReClor and LogiQA are in $[3, 6)$ and $[0, 3)$, respectively.

Comparing the accuracies of RoBERTa-large baseline, prior SOTA DAGN and our proposed FOCAL REASONER in Figure 6, our model outperforms baseline models on all the divided subsets, which demonstrates the effectiveness and robustness of our proposed method. Specifically, for ReClor, FOCAL REASONER performers better when there are more fact units in the context, while for LogiQA, FOCAL REASONER works better when the number of fact units locates in $[0, 3)$ and $[9, 12)$. The

| Number | ReClor | | LogiQA | |
|---|---|---|---|---|
| | Train | Dev | Train | Dev |
| Fact Unit Argument | 14,895 | 1,665 | 20,676 | 1,981 |
| Named Entity | 9,495 | 984 | 12,439 | 1,515 |

Table 4: Statistics for fact unit entities and traditional named entities in datasets.

| Model | S | W | I | CMP | ER | P | D | R | IF | MS |
|---|---|---|---|---|---|---|---|---|---|---|
| RoBERTa$_{large}$ [7] | 61.70 | 47.79 | 39.13 | 63.89 | 58.33 | 50.77 | 50.00 | 56.25 | 61.54 | 56.67 |
| DAGN [10] | 63.83 | 46.02 | 39.13 | 69.44 | 57.14 | 53 85 | 46.67 | 62.50 | 62.39 | 56.67 |
| FOCAL REASONER | 65.96 | 51.33 | 43.48 | 72.22 | 67.86 | 53.85 | 50.00 | 62.50 | 62.39 | 60.0 |

Table 5: Accuracy on the dev set of ReClor corresponding to several representative question types. *S: Strengthen, W: Weaken, I: Implication, CMP: Conclusion/Main Point, ER: Explain or Resolve, D: Dispute, R: Role, IF: Identify a Flaw, MS: Match Structures.*

reason may lie in the difference in style of the two datasets. However, all the models include ours struggle when the number of fact units is above certain thresholds, i.e., the logical structure is more complicated, calling for better mechanisms to cope with.

### 5.3 Interpretability: a Case Study

We aim to interpret FOCAL REASONER's reasoning process by analyzing the node-to-node attention weights induced in the supergraph in Figure 7. We can see that our FOCAL REASONER can well bridge the reasoning process between context, question and option. Specifically, in the graph, "students rank 30%" attends strongly to "playing improve performance". Under the guidance of question

| Dataset | $[0, 3)$ | $[3, 6)$ | $[6, 9)$ | $[9, 12)$ | $[12, \infty)$ |
|---|---|---|---|---|---|
| ReClor | 37.2% | 48.6% | 12.6% | 0.6% | 1.2% |
| LogiQA | 47.5% | 37.5% | 10.9% | 3.5% | 0.6% |

Table 6: Distribution of fact unit number on dev set of the training datasets.

to select the option that weakens the statement and option interaction, our model is able to tell that "students rank 30% can play" mostly undermines the conclusion that "playing improves performance".

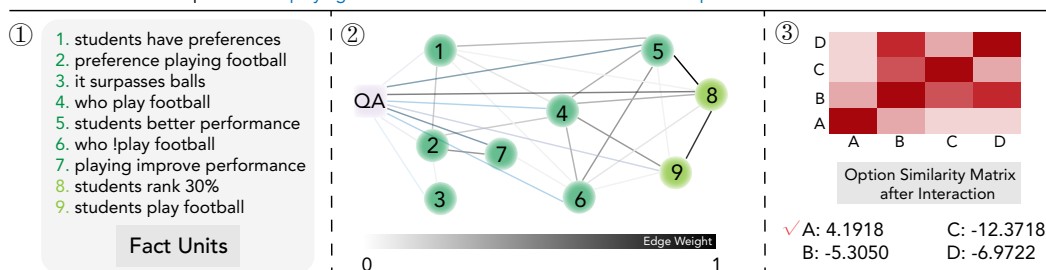

Figure 7: An example of how our model reasons to get the final answer.

## 6 Conclusion

For logical reasoning arising from machine reading comprehension, it is well known that clear and accurate forms like global knowledge are crucial. In this work, we make a finding that existing studies miss focusing on quite a lot of non-knowledge parts which is also indispensable for better reasoning. Thus we propose extracting a general form called "fact unit" to cover both global and local logical units, hoping to shed light on the basis of structural modeling for logical reasoning. Our proposed FOCAL REASONER not only better uncovers the logical structures within the context, which can be a general method for other sophisticated reasoning tasks, but also better captures the logical interactions between context and options. The experimental results verify the effectiveness of our method.

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
