# OpenReview forum: "Fact-driven Logical Reasoning"
_NeurIPS.cc/2021/Conference — NeurIPS 2021 Submitted_

### Official Review · Reviewer_Bigy · 2021-07-05

**Rating:** 5
**Confidence:** 4

**Summary:**

This paper presents a graph-based approach to solve logical reasoning questions. This method first extracts fact units from given questions, passages and answer options and builds a graph based on them, where a dependency parser is used to extract <subject, verb, object> triples. Then, the pre-trained RoBERTa large is used to generate representations of the concatenated question-passage-option sequences. Next, a graph is built based on the dependency parsing results of the sequence, and a graph attention network is used to perform neural message passing to exchange information between graph nodes. Last, all information are aggregated and used to do the prediction. Evaluations are conducted on 3 reasoning tasks, ReClor, LogiQA and MuTual, with ablation study performed for the effects of global atoms, fact unit variants/numbers and interaction layers.

**Ethical Concerns:**

no concern

**Ethics Review Area:**

["I don’t know"]

**Limitations And Societal Impact:**

I don’t think this work has any negative societal impact.

**Main Review:**

The key contribution is to leverage the global and local knowledge to solve logical reasoning tasks. But based on my understanding, this approach is not new. For example, in the commonsense reasoning work https://arxiv.org/abs/1909.05311, a method was proposed to use graphs and graph neural networks for the commonsense qa task. Although there are some details different, such as the proposed work uses dependency parsing instead of semantic role labeling, uses given passage so it doesn’t need to retrieval external knowledge as evidences, it still shares a similar architecture with the previous work. Besides, for the ReClor task, the latest benchmark is https://arxiv.org/abs/2105.03659, I suggest the authors to include this paper as a reference and compare with it.

**Time Spent Reviewing:**

2 hour

---

> ### Author Response · Authors · 2021-08-06
> **Response to Reviewer Bigy**
>
> Thanks for your time and comments, and the following are our responses.
>
> 1. About the commonsense reasoning work (novelty).
>
>     The key contribution, as you have pointed out, is the proposal of fact unit as the natural reasoning basis, which captures both global and local knowledge, not the model architecture. Compared with the mentioned commonsense reasoning work, the major differences are as follows:
>     - The task setting is different. For commonsense reasoning, they require external knowledge such as Concept Net to help, which is non-existent in original content. In our work, we try to capture the existing knowledge via a general fact unit to uncover the logical chains and aid reasoning.
>     - Our motivation is quite different. For commonsense reasoning, they only focus on global knowledge (from knowledge graph), relying on the information of named entities/semantic roles and the corresponding relations. Instead, our work constructs a general and meaningful reasoning basis, which is a superset of their work.
>     - In the technical aspect, to construct the fact unit, we use self/reverse-in/default-in, etc. edges to capture the relations inside the fact unit, and further organize them into super-graphs to model the relations among fact units. While in their work, they only use plain text representations to construct concept-graph.
>
>     Our work provides comprehensive, in-depth, and empirical experiments (in Section 5.1) into what the super-graphs are constituted and how the components can be organized in a general view. Experiments demonstrate that our proposed fact units are a better representation of knowledge for logical reasoning and the way we organize them into supergraphs also contributes.
>
>
> 2. Comparison with the latest benchmark
>
>     Please note that this is a contemporary work (LReasoner) with ours. The preprint date is 5.8, which is very close to the NeurIPS deadline on 5.28. We will add that in our later version for comparison.
>
>     Our model achieves comparable performance with LReasoner. It employs symbolic reasoning and data augmentation techniques, which is in a different research line from ours. Without data argumentation, LReasoner shows relatively weaker results, showing that our fact-driven approach would be beneficial compared with the symbolic-driven technique.
>
>       | Model               | ReClor-Dev | ReClor-Test | LogiQA-Dev | LogiQA-Test |
>       | ------------------- | :----------: | :-----------: | :----------: | :-----------: |
>       | RoBERTa-large       | 62.6       | 55.6        | 35.0       | 35.3        |
>       | LReasoner           | 66.2       | 62.4        | -          | -           |
>       |     - data augmentation | 65.2       | 58.3        | -          | -           |
>       | Ours                | 66.8       | 58.9        | 41.0       | 40.3        |
>
>     Compared with the neural methods for logical reasoning, symbolic approaches would rely heavily on dataset-related predefined patterns which entail massive manual labor, potentially restricting the generalizability of models.

---

> > ### Comment · Reviewer_Bigy · 2021-08-23
> > **thank you for the explanation**
> >
> > Thank you for the answers to my questions and concerns! I also read your feedbacks to questions from other reviewers. However, I stick to my previous decision, and please DO include all these good insights and explanations in the next version of the paper.

---

### Official Review · Reviewer_hoPP · 2021-07-16

**Rating:** 5
**Confidence:** 4

**Summary:**

This paper presents a system for graph-based reasoning about scenarios. Facts are represented as subgraphs which are connected into a supergraph via a single "global" node and coreference nodes. A passage, question, and options are first passed through RoBERTa, then a neural network (graph attention network) passes information through the graph representation before an answer is predicted. Results show that this model works better than a related graph-based technique (the DAGN model), with particularly strong gains on the LogiQA test set.

**Limitations And Societal Impact:**

This has been addressed adequately.

**Main Review:**

The core idea of this paper is a pretty nice one. Merging global and local (instance-specific) information is a great one and is very well-motivated for improving machines' ability to reason. However,
I don't actually see where the notion of global (cross-instance facts) is integrated into the paper? What I expected to see was local information from, say, a ReCLOR example combined with
some retrieved commonsense. The global node in the graph just seems to unify all the information in a single example.

Therefore, I will judge the paper mostly in terms of its architecture for doing local reasoning over a single example.

Originality: The ideas here are original, although much of the neural machinery is known. In my view, a weakness of this paper is actually the sophistication of the neural model: many small changes to the neural architecture (compared to pre-trained models) make it a bit harder to attribute the gains to specific aspects of the approach, although the paper's ablations are well done.

Quality: The work is technically sound as far as I can tell. The methods used are appropriate and the results are reasonably in-depth, including ablations of the various parts of the model.

Clarity: The paper is well-written overall.

Significance: This is where I am most conflicted about the paper.  The results are difficult to interpret. Certainly there are gains from the paper's technique and the ablations show that the different components of the model all contribute to the performance. But when fairly important elements of the model like coreference edges contribute only around 2% accuracy, it's hard to know how much to trust the narrative here about reasoning.

More generally, in a multiple-choice task, it's hard to know that the model is really behaving as advertised rather than just adding capacity on top of the baselines. It's not easy to form an apples-to-apples comparison in terms of number of parameters. The case study in Section 5.3 shows that something is happening in the graph component, but whether this is causally associated with the ability to get the right answer as opposed to a byproduct is hard to determine.

The authors may wish to cite Mrinmaya Sachan's work on using representations from AMR: https://www.aclweb.org/anthology/P16-2079.pdf

The LogiQA results show some nice improvements, but I'm curious about the statistical significance statement in Table 1: are all of these results (including ReCLOR) truly representing significant gains? Compared to which baseline(s)?

Overall, I'm borderline about this paper. There are some nice core ideas but I don't know whether these ideas of how to use graphs are proven to be significantly better than prior work. The complexity of the work isn't necessarily justified by strong results on ReCLOR.

**Time Spent Reviewing:**

1.5

---

> ### Author Response · Authors · 2021-08-06
> **Response to Reviewer hoPP**
>
> Thanks for your time and insightful comments.
>
> The core idea of the paper is to use fact unit, which is quite simple to extract and can well represent global and local knowledge. Based on this intuition, we design the model architecture to model such knowledge.
>
> 1. Notion of global knowledge.
>
>     We have provided an example in Figure 3, demonstrating the global knowledge extracted by existing methods. Compared with it, our proposed fact units can cover both global and local knowledge. Named entity is one such way reflecting the global knowledge, which is implemented and compared with our proposed methods in Section 5.1 “Fact unit variants”. Yes, we will add a clearer example to demonstrate the notions. Global node is just a way to organize the extracted fact units, it is different from the “global knowledge” concept.
>
>
> 2. The sophistication of the neural model & interpretation of the results.
>
>     The major contribution of our paper is not simply the model architecture. Please see the **General Response** for more details.
>     - Firstly, the number of parameters of our model is around 15% increase of the original RoBERTa-large model, which is on par with DAGN. Given the necessary part of graph neural network, our model architecture is not complicated as thought.
>     - Secondly, from our ablation studies in Section 5.1, we can see that the origin of fact unit (Fact Unit Variants) and how to organize them (co-ref, global node, edge type) contribute the majority of the performance. (2% for co-ref is not a minor number, given the fact that co-ref is just one of the three strategies)
>
>
> 3. Significance of the results.
>
>     Yes, the dev and test results are significant with p-value < 0.05 based on the t-paired test. The compared baseline model is DAGN.

---

> > ### Comment · Reviewer_hoPP · 2021-08-17
> > **Thanks for the rebuttal**
> >
> > I appreciate the response and the general response. However, I think this paper still needs another round of revision along the dimensions we've discussed here before being ready for acceptance.

---

### Official Review · Reviewer_Tmgr · 2021-07-18

**Rating:** 4
**Confidence:** 4

**Summary:**

The paper claims that a lot of previous work in QA-based MRC focused too much on entity-aware common sense knowledge and other clues in the natural language (in passage, questions & candidate options). To overcome this, the authors extract "fact units" in the form of "subject-verb-object" from syntactic processing (dependency parsing).  With these fact units as a node, they form a supergraph and uses GNNs to process this. There are also many more components: global node, logical fact regularization, enlarged edge types compared to previous work and interactions. In the experiment section, they show that FOCAL REASONER outperforms the previous SOTA models and run an ablation study on how each component contributes (in Table 3).

**Limitations And Societal Impact:**

The author did not address "limitations and potential negative societal impact of their work".
Although this paper's main focus is in the reasoning from MRC, as it is merely learning from a supervised model, would this model be robust to social biases in the training texts?

**Main Review:**

[Originality/Quality]

The strength of this paper is their extensive experiments and good performance which beats the previous SOTA performance.

However, given the complex system that the paper works on, I wasn't sure what this paper's main focus is nor what their scientific contribution/novelty is.

After reading this paper back-and-forth, the claimed main contribution seems to be bringing 'fact" from "syntactic processing" to bring "local complete facts". But I do not feel like the main story of the paper is revolving around this contribution of "local fact extraction" and most of the paper discusses the complex model structure.
The part that really discusses how different fact units impact the system is only discussed in section 5.1 "Fact unit variants" (233-251) and how they syntactically extract facts are in 115-118. And even if this is the main contribution of this paper, I am not sure if this contribution is novel enough to be accepted as a full paper in NeurIPS.

Given many of the complex model structures this paper has, and as many of them are borrowed from previous work (using GNN and  logical fact regularization), I think authors should set their stance on what their contribution exactly is.  For example, if the paper don't use logical fact regularization loss, the performance drops from 66.8 --> 64.2 which is lower than competing models.
Why all these components are required to claim that local syntactic facts are really important was not clear to me. Also, I wasn't sure what information to gain from the comparisons with previous models in Table 1, 2. What was lacking in Table 1,2 for me was answers to the following quesitons:
* Are the baseline models only "global" information?
* Is FOCAL REASONER just having better results because it has different model structure?
* Ultimately, what information are the baseline models providing in order to satisfy the claims this paper makes?

One recommendation to authors is, rather than bringing best of everything, trying to create more comparable baselines that can justify the claim (such as analysis on "Fact unit variants")
Perhaps, if this paper's real goal was improving SOTA by novel model structures, I think it should focus more on what is the new wheel they are bringing into this framework.

[Clarity]
The paper needs to be written more clearly. I found many parts of the definitions to be ambiguous as can be seen in “question” section of the review.

[Corrections]
* Figure 1. Don’t see the “relations” in green, maybe you meant relations in orange?
* 234-235:  there are another two ways  in different granularities for construction—> there are two other ways to construct ...

[Questions]
* I am not sure what definition 1 brings (From triplet to fact unit, it seems the same to me? Also, the fact unit is defined as the triplet on the cited paper [13] (as fact candidate in [13]).
* Definitions are confusing as there are overloaded concepts here and there. It may be useful to have examples for each of them (although authors try to depict examples in the figure).
e.g. 120-123 Since V,E,R are overloaded between fact units and Levi graph, it might be nice to have examples to distinguish, V,R of F and V_L of F_L.
* Figure 4 edge for “fact-driven reasoning process” box only relies on question & option vectors, is this correct? (Although they are contextualized by the passage, I thought the passage is the main source of the super graph as shown in figure 5?)
* How can fact units be replaced with named entities when they don’t have any relations?
* In Table 4, did you replace fact units to be SRL output? And used it in your current Focal Reasoner with all the E ={same, cored, global} and edges R_L={ default-in, default-out, reverse-in, reverse-out, self} ?



**Time Spent Reviewing:**

5

---

> ### Author Response · Authors · 2021-08-06
> **Response to Reviewer Tmgr**
>
> Thanks for your devotion and the insightful reviews.
>
> 1. Are the baseline models only "global" information?
>
>     Yes, previous methods commonly focus on the reasoning based on the relationships between named entities and semantic roles, which are defined as global facts according to Figure 2. In contrast, our method is general by considering both global and local facts as supergraphs to provide a comprehensive basis for logical reasoning. We have compared the performance with the variants using named entities and semantic roles in Table 3 (Fact Unit Variants).
>
>
> 2. Is FOCAL REASONER just having better results because it has a different model structure?
>
>     Not exactly, the better results would attribute to our fact-driven supergraph modeling (according to Table 3), which essentially motivates our model design as a holistic architecture. Please see the **General Response** for detailed clarification.
>
>     We will improve the presentation to clearly show the contribution of the main parts of our model and compare different alternatives.
>
>
> 3. Ultimately, what information are the baseline models providing in order to satisfy the claims this paper makes?
>
>     Our claims are supported by the ablations in Table 3, where each of the variants can serve as such a baseline. In a broader scope, the RoBERTa baseline is used to verify the effectiveness of our holistic architecture. And the previous SOTA model DAGN can verify the necessity of modeling both the global and local facts.
>
>
> 4. What does definition 1 brings.
>
>     Fact unit is the combination of the triplet and the relations (e.g., E1 -> R, R -> E2) in the triplet.
>
>
> 5. Overloaded concepts.
>
>     We will improve the representation and add more clear examples following your suggestions.
>
>
> 6. Figure 4 edge for “fact-driven reasoning process” box only relies on question & option vectors, is this correct?
>
>     The main source of “fact-driven reasoning process” is the contextualized information of passage. We will improve the representation of Figure 4.
>
>
>
> 7. How can fact units be replaced with named entities when they don’t have any relations?
>
>     Those relations are obtained based on the original fact unit. Specifically, we first extract all the named entities with StanfordCoreNLP, we discard the parts and their corresponding relations that are not in the extracted named entities. In other words, we only save the subgraph that is related to the named entities.
>
>
> 8. Issues regarding SRL replacement.
>
>     Yes, we replace fact units with SRL tagging generated via AllenNLP. The other settings are the same as fact units.

---

> > ### Comment · Reviewer_Tmgr · 2021-08-20
> > **Response to the rebuttal**
> >
> > I appreciate the prompt responses the authors have made. However, at the state the paper was initially submitted, I don't think small clarifications and comments can bring this paper to acceptance after rebuttal. I agree with reviewer hoPP that the paper is not ready yet and would become a stronger candidate after reflecting reviewer's comments.

---

### Author Response · Authors · 2021-08-06
**General Response**

Dear reviewers,

Thanks so much for your time and the constructive reviews. We also appreciate your recognition of the core idea, motivation, extensive experiments, and good performance. We have studied all the comments carefully and have prepared point-to-point feedback. Firstly, we would like to address the general concern about the “complex model structure”.

Our contribution is the motivation of merging global and local knowledge for logical reasoning. Based on such motivation, the adopted graph modeling is a must-be solution. Although we did not explicitly discuss the model complexity in the submission, our goal is not stacking complex modules:

1)	We build supergraphs on top of the fact units from syntactic processing, to capture both global connections between facts. To model the supergraph, a natural solution is using GNN.
2)	Since the arguments and predicate should be closely related with some explicit relationships, the logical fact regularization technique is designed to help model the local concepts or actions inside a fact.
3)	For the application to the concerned QA tasks that require reasoning, an intuitive solution is to model the interaction between the question and options.

Based on the motivation and the resulting structure above, we achieve substantial improvements in the model performance. Although Table 3 has presented a comprehensive comparison of the module selections, our method is not supposed to be complex. Some of the modules are dispensable (e.g., the  QA context node), our original intuition is just to provide readers with the comparison between any possible alternative components as much as possible.

For the model complexity, our method basically keeps as simple as previous models like DAGN. Our model only has 414M parameters, where the current SOTA (on RoBERTa) also has about 395M. We both employ GCN to model the graph-like relationships, which result in the major increase of parameters compared with the RoBERTa baseline (355M).

We will improve the paper presentation accordingly.

---

### Author Response · Authors · 2021-08-17
**General Response Update**

We thank all reviewers so much for the valuable comments on improving the quality of this work. We have revised the paper according to the feedback and our latest evaluations. The major revisions are listed as following:
1)	We reorganize the introduction part to bring out the best of Fact-driven reasoning and the relevant concepts, e.g., Fact Unit.
2)	We improve Figure 4 and Figure 5 to help with the understanding of concepts.
3)	We add a discussion to highlight the Fact Unit's contribution against other prevalent methods that only use global knowledge in Section 5.
4)	We rewrite Section 3 methodology to make the narration of the model best fit with the motivation, i.e., merging global and local knowledge.
5)	We detail the main performance analysis in Section 4.3, where we add the latest benchmark LReasoner as one of the baseline models for comparison in Table 1. We also analyze the model complexity in terms of trainable parameters with baselines.

The above updates are already done, and we will upload the revised version when we have permission.

---

### Author Response · Authors · 2021-08-26
**General Response**

Dear reviewers,

We appreciate your update.

We have made all the required revisions accordingly and the paper actually has been well updated since we expected this NeurIPS which adopts open review system like ICLR may allow to update the revised paper itself. However, it is a big pity that the review mechanism of this NeurIPS does not do so. We understand most of review comments just expect such a revised version of our paper. We promise such a required paper has been existing but cannot submit anyway due to victim of the current review mechanism setting.

The major concern of the current version is the presentation of the idea and method proposed (which one to focus on). We introduce the model in quite a detail, which does not mean that our model is complex. We just want to keep reproducibility of this work as much as possible.

According to the content of the discussion, we only need to explain the core contribution and simplify the introduction of the method to update the article, which is easy to solve, but the fact is that we have revised the article long ago, but we cannot submit the update.

Finally, thank you very much for your review.

---

### Decision · Program_Chairs · 2021-09-27

**Decision:**

Reject

**Comment:**

This paper proposes to extract subgraphs containing local factual information from the given QA context (in the form of subject-predicate-object triples), connect them via a single global node and coreference, and perform graph-based reasoning over the resulting structure. The reviewers note the main strength of the work to be its good empirical results on the 3 benchmarks considered. However, given the complex nature of the proposed system, it is difficult to understand which aspects of the proposed technique are truly effective.

For instance, I see from Table 1 that the gain on ReClor and LogiQA "test" sets is only about 1 percentage point, compared to the best existing baseline (DGAN). While this gain may be statistically significant (i.e., not a result of random chance) because the test set is large enough, it is difficult to rely mainly on a 1 point gain to justify a new approach.

The active response of the authors is much appreciated. It would certainly make the paper much stronger once the explanations and new results from the response are included in the next draft of the paper. I'm sorry the NeurIPS reviewing model does not allow uploading a revised version of the paper (similar to many conferences). The changes needed before publication, however, are substantial and would generally merit a fresh round of review.